# CHUNKRAG: A NOVEL LLM-CHUNK FILTERING METHOD FOR RAG SYSTEMS

## ABSTRACT

Retrieval-Augmented Generation (RAG) frameworks leveraging large language models (LLMs) frequently retrieve extraneous or weakly relevant information, leading to factual inaccuracies and hallucinations in generated responses. Existing document-level retrieval approaches lack sufficient granularity to effectively filter non-essential content. This paper introduces ChunkRAG, a retrieval framework that refines information selection through semantic chunking and chunk-level evaluation. ChunkRAG applies a dynamic greedy chunk aggregation strategy to segment documents into semantically coherent, variable-length sections based on cosine similarity. Empirical evaluations on the PopQA, PubHealth and Biography dataset indicate that ChunkRAG improves response accuracy over state-of-the-art RAG methods. The analysis further demonstrates that chunk-level filtering reduces redundant and weakly related information, enhancing the factual consistency of responses. By incorporating fine-grained retrieval mechanisms, ChunkRAG provides a scalable and domain-agnostic approach to mitigate hallucinations in knowledge-intensive tasks such as fact-checking and multi-hop reasoning.

## 1 INTRODUCTION

LLMs combined with retrieval-augmented generation (RAG) have improved AI systems' ability to generate informed responses using external knowledge. While promising for knowledge-intensive tasks, RAG systems often struggle with retrieving irrelevant or weakly relevant content. Despite using techniques like re-ranking and query rewriting, this limitation leads to factual errors and hallucinations in the generated outputs.

Current RAG systems often retrieve large document segments, assuming more content means better coverage. However, this overlooks the need to evaluate smaller sections independently, leading to the inclusion of irrelevant information. LLMs' inability to verify factual accuracy compounds this issue, reducing RAG reliability in applications like question answering and decision-making(Ji et al., 2023; Min et al., 2023a).

Figure 1 illustrates the impact of chunk filtering on response generation. Without chunk filtering (top), irrelevant information, such as references to other French cities, is incorporated into the response. In contrast, LLM-driven chunk filtering (bottom) removes unnecessary content, yielding a precise response: *"The capital of France is Paris."* Recent approaches like CRAG and Self-RAG (Your et al., 2024; Asai et al., 2024) have tried to improve retrieval accuracy through corrective retrieval and self-reflection mechanisms. However, these methods still operate at the document level, failing to adequately filter individual text chunks (Shi et al., 2023). This granularity issue leaves RAG systems susceptible to misleading information.

We propose ChunkRAG, a novel approach of LLM-driven chunk filtering. This framework operates at a finer level of granularity than traditional systems by supporting chunk-level filtering of retrieved information. Rather than determining the relevance of entire documents, our framework evaluates both the user query and the individual chunks within the retrieved chunks. The large language model assesses the semantic relevance of each chunk in relation to the user's query, thereby enabling the system to filter out irrelevant or weakly related chunks before they reach the generation stage.This approach shows particular promise for knowledge-intensive tasks, such as multi-hop reasoning and fact-checking (Piktus et al., 2021; Rony et al., 2022).

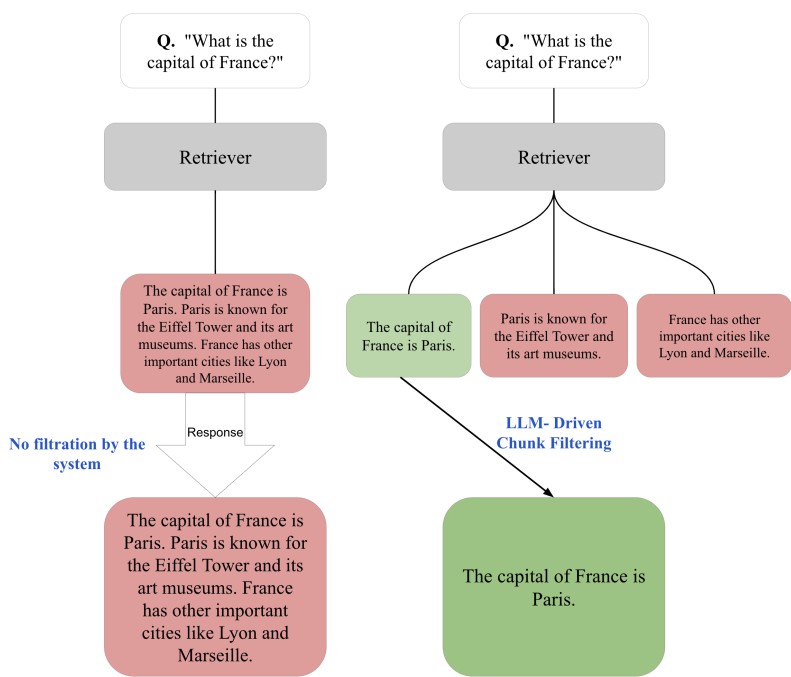

Figure 1: Comparison of Response Generation With and Without Chunk Filtering

## 2 RELATED WORKS

### 2.1 ADDRESSING HALLUCINATIONS IN LARGE LANGUAGE MODELS

Large language models (LLMs) have made substantial progress in instruction understanding and text generation (Bang et al., 2023; Qin et al., 2023; Zhong et al., 2023). Nevertheless, they continue to suffer from hallucinations—outputs that are incorrect. Research suggests that erroneous internal knowledge often triggers these hallucinations (Tonmoy et al., 2024; Zhang et al., 2023b; Shuster et al., 2021), aggravated by low-quality data distributions in training as well as a lack of built-in verification mechanisms. Consequently, improving access to high-quality external knowledge remains an important line of research.

### 2.2 RETRIEVAL-AUGMENTED GENERATION TECHNIQUES

Retrieval-Augmented Generation (RAG) has gained traction as an effective strategy to mitigate hallucinations (Lewis et al., 2020; Guu et al., 2020). RAG systems enhance performance on knowledge-intensive tasks by injecting relevant retrieved documents during generation. However, the quality of outputs depends heavily on retrieval accuracy, as poor document selection can increase factual errors. Recent work has explored ways to refine RAG pipelines to better filter irrelevant context (Kim et al., 2024; Wang et al., 2024; Liu et al., 2024). For example, (Asai et al., 2024) introduced Self-RAG, which integrates a "critic" mechanism to determine when retrieval is truly necessary. (Your et al., 2024) proposed CRAG, augmenting RAG with strategies to correct weakly appurtenant, unsubstantiated retrieval results. Similarly, (Yoran et al., 2024) leveraged a Natural Language Inference (NLI) model to filter out irrelevant contexts, leading to more robust systems. (Smith et al., 2023) introduced Multi-Meta-RAG, which improves multi-hop reasoning by using LLMs to extract metadata for more effective database filtering before retrieval. This metadata-driven approach helps combine relevant context from different domains while reducing noise, ultimately leading to more coherent responses.

## 2.3 QUERY REWRITING FOR ENHANCED RETRIEVAL

A key challenge is bridging natural language queries with document storage formats. (Johnson & Lee, 2023) proposed a "Rewrite-Retrieve-Read" framework where a trainable query rewriter transforms user queries into forms that better match corpus content. By incorporating relevant keywords and domain terms, this approach improves passage retrieval accuracy(Ma et al., 2023). The rewriter is optimized through reinforcement learning based on question-answering performance(Liu & Mozafari, 2024). Through such automated query rewriting, retrieval modules can better capture relevant documents, especially for queries that use informal language or lack domain-specific keywords(Li et al., 2024; Mao et al., 2024).

## 2.4 REDUNDANCY REDUCTION WITH COSINE SIMILARITY

Redundant information in retrieved documents can clutter context. Using cosine similarity, near-identical sections can be deduplicated by filtering chunks exceeding a similarity threshold (e.g., $> 0.9$) (Liu et al., 2023), streamlining input and reducing confusion from repetition.

## 3 METHODOLOGY

The core objective of this work is to mitigate hallucinations and irrelevant responses generated by Retrieval-Augmented Generation (RAG) systems. Our proposed methodology follows a two-stage approach: semantic chunking and advanced filtering to refine retrieval results.

### SEMANTIC CHUNKING

Semantic chunking serves as the foundational step of our methodology, transforming the input document into semantically meaningful units to facilitate effective retrieval. This stage involves three sub-processes:

- **Input Preparation**: We begin by tokenizing a document $D$ into sentences using NLTK's *sent_tokenize* function. Each sentence is then assigned an embedding vector, generated using a pre-trained embedding model (`text-embedding-3-small`).

- **Chunk Formation**: Consecutive sentences are grouped into chunks based on their semantic similarity, measured by cosine similarity. Specifically, if the similarity between consecutive sentences drops below a threshold ($\theta = 0.8$), a new chunk is created, as this indicates a shift to a different subtopic or theme that warrants its own grouping. Each chunk is also further constrained to be under 500 characters to enable granular search and prevent oversized chunks - even when discussing a single topic, very large chunks can hinder precise information retrieval during tasks like question answering. This character limit ensures efficiency during subsequent stages.

- **Chunk Embeddings**: Each chunk is represented using the same pre-trained embedding model as above. The resultant chunk embeddings are stored in a vector database to facilitate efficient retrieval during the query phase.

### HYBRID RETRIEVAL AND ADVANCED FILTERING

In the retrieval and filtering phase, we integrate conventional RAG components with advanced fine-tuning techniques to better retrieval.

- **Retriever Initialization and Query Rewriting**: We initialize a retriever capable of comparing user queries against the chunk embeddings. To enhance query efficacy, we apply a query rewriting step using GPT-4o. It adapts user inputs into forms better aligned with chunk embeddings.

- **Initial Filtering**: Retrieved chunks are initially filtered using a combination of TF-IDF scoring and cosine similarity. Chunks with high redundancy (similarity $> 0.9$) are eliminated. The remaining chunks are sorted based on their similarity to the rewritten query.

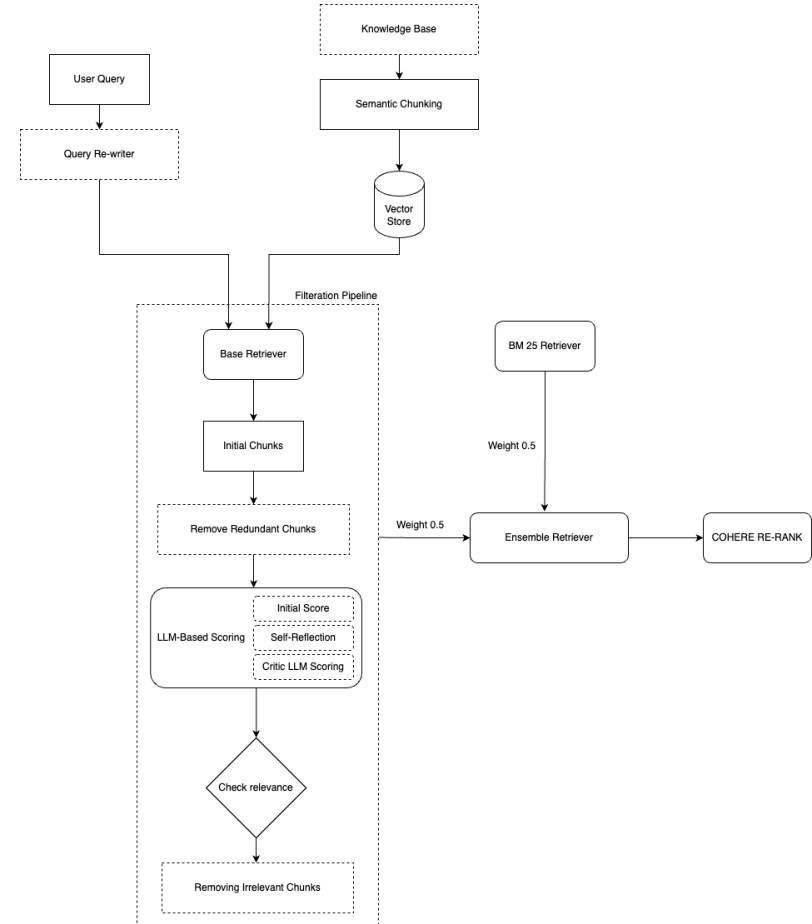

Figure 2: ChunkRAG Methodology for Enhanced Retrieval and Filtering. This figure illustrates the ChunkRAG pipeline, combining semantic chunking, filtering, and ensemble retrieval to optimize information relevance and accuracy, with final results re-ranked for precision.

- **Relevance Scoring and Tresholding**: Each chunk's relevance is evaluated through a multi-stage process: an LLM assigns initial scores, followed by self-reflection and critic model refinements. The self-reflection step assesses query alignment, while the critic applies domain-specific heuristics (e.g., temporal consistency for time-sensitive queries). A dynamic threshold, based on score distribution analysis, determines final chunk selection. When scores cluster tightly, the threshold increases to retain only the most relevant chunks.

- **Hybrid Retrieval Strategy**: We combine BM25 and LLM-based retrieval methods with equal weights (0.5 each) to balance keyword and semantic matching. Cohere's reranking model (`rerank-englishv3.0`) then addresses the **Lost in the middle** problem - where relevant information in the middle of long documents tends to be underemphasized by standard retrieval methods - by re-evaluating chunks with emphasis on contextual centrality, preventing the oversight of relevant mid-document information.

RESPONSE GENERATION AND EVALUATION

After filtering, the remaining chunks are used as context to generate the final response. The steps include:

- **Response Generation**: An LLM generates a response based on the filtered context chunks. During generation, strict constraints ensure that only retrieved information is used, thereby minimizing the risk of hallucinations.

---

**Algorithm 1** Enhanced Hybrid Retrieval and Filtering

---

**Require:** $q$: Original user query
**Require:** $\mathcal{D}$: Document collection
**Require:** $\lambda_{dup}$: Redundancy threshold (e.g., 0.9)
**Require:** $w_{bm25}, w_{llm}$: Hybrid retrieval weights
**Ensure:** $\mathcal{C}_{final}$: Filtered and ranked chunks
1: $q_{rewritten} \leftarrow$ GPT4_QueryRewrite($q$)
2: **// Hybrid Retrieval**
3: $\mathcal{C} \leftarrow$ CombineRetrieval(BM25($\mathcal{D}, q_{rewritten}$), LLM($\mathcal{D}, q_{rewritten}$), $w_{bm25}, w_{llm}$)
4: **// Redundancy Removal**
5: $\mathcal{C}_{filtered} \leftarrow \varnothing$
6: **for** each chunk $c_i \in \mathcal{C}$ **do**
7:      **if** $\max\limits_{c_j \in \mathcal{C}_{filtered}} \cos\big(\text{emb}(c_i), \text{emb}(c_j)\big) \leq \lambda_{dup}$ **then**
8:          Append $c_i$ to $\mathcal{C}_{filtered}$
9:      **end if**
10: **end for**
11: **// Multi-stage Scoring**
12: **for** each chunk $c \in \mathcal{C}_{filtered}$ **do**
13:      $base \leftarrow$ LLMRelevance($c, q_{rewritten}$)
14:      $reflect \leftarrow$ SelfReflect($c, q_{rewritten}, base$)
15:      $critic \leftarrow$ CriticEval($c, q_{rewritten}, base, reflect$)
16:      $score(c) \leftarrow$ CombineScores($base, reflect, critic$)
17: **end for**
18: **// Dynamic Thresholding**
19: $S \leftarrow \{ score(c) \mid c \in \mathcal{C}_{filtered} \}$
20: $\mu \leftarrow$ mean($S$); $\sigma \leftarrow$ std($S$)
21: $T \leftarrow$ if var($S$) $< \epsilon$ then $\mu + \sigma$ else $\mu$
22: $\mathcal{C}_{threshold} \leftarrow \{ c \in \mathcal{C}_{filtered} \mid score(c) \geq T \}$
23: **// Lost-in-Middle Reranking**
24: $\mathcal{C}_{final} \leftarrow$ Cohere_Rerank($\mathcal{C}_{threshold}, q_{rewritten}$)
25: **return** $\mathcal{C}_{final}$

---

- **Evaluation**: The generated responses are evaluated for accuracy against a set of human-validated reference answers.

Our methodology(Figure 2 and Algorithm 1), combining semantic chunking with advanced retrieval and filtering mechanisms, significantly enhances the quality of responses produced by RAG systems, ensuring both relevance and correctness of the generated content.

# 4 EXPERIMENTS

The experiments were conducted on the free-tier Google Colab environment, which provides a standard NVIDIA K80 GPU with 12GB of memory. As such, due to computational resource constraints, our evaluation was primarily focused on the PopQA, PubHealth and Biography dataset.

## 4.1 TASKS, DATASETS AND METRICS

ChunkRAG was evaluated on three datasets, which are in public domain and licensed for research purposes, including:

**PopQA** (Mallen et al., 2023) is a *short*-form generation task. Generally, only one entity of factual knowledge is expected to be answered for each single question. In our experiments, we exactly followed the setting in Self-RAG (Asai et al., 2024) which evaluated methods on a long-tail subset consisting of 1,399 rare entity queries whose monthly Wikipedia page views are less than 100. Accuracy was adopted as the evaluation metric.

**Biography** (Min et al., 2023b) is a *long*-form generation task that is tasked to generate a detailed biography about a certain entity. Following previous work, FactScore (Min et al., 2023b) was adopted to evaluate the generated biographies.

**PubHealth** (Zhang et al., 2023a) is a task in health care domain consisting of true-or-false questions. Claims are represented about health with factual information, and the model is tasked to verify the authenticity and give the judgment. Accuracy was adopted as the evaluation metric.

## 4.2 BASELINES

### 4.2.1 BASELINES WITHOUT RETRIEVAL

We first evaluated several models that do not incorporate any retrieval mechanisms. Among the public LLMs, we included LLaMA2-7B and LLaMA2-13B (Touvron et al., 2023), known for their versatility across diverse natural language processing (NLP) tasks, and Alpaca-7B and Alpaca-13B (Dubois et al., 2023), which are instruction-tuned models optimized for effectively following user prompts. For proprietary models, we included LLaMA2-chat13B, a conversational variant of LLaMA2 tailored for dialogue-based applications, and ChatGPT, OpenAI's proprietary conversational agent renowned for its robust language understanding and generation capabilities. These baseline results are taken from (Your et al., 2024).

### 4.2.2 BASELINES WITH RETRIEVAL

**Standard Retrieval-Augmented Generation (RAG):** To establish a baseline for retrieval-augmented methods, we evaluated standard RAG approaches. Specifically, we employed Standard RAG (Lewis et al., 2020), which utilizes a retriever to fetch relevant documents based on the input query, subsequently feeding these documents into the language model to generate responses. For consistency, we utilized the same retriever mechanism as ChunkRAG to ensure a fair comparison. In addition to Standard RAG, we evaluated instruction-tuned LLMs with standard RAG, including LLaMA2-7B, LLaMA2-13B, and Alpaca-7B, Alpaca-13B, to assess the impact of retrieval augmentation in conjunction of instruction tuning. These baseline results are taken from (Your et al., 2024).

**Advanced Retrieval-Augmented Generation:** To benchmark ChunkRAG against more sophisticated RAG-based methods, we included advanced systems that incorporate additional strategies to enhance performance. Self-RAG (Asai et al., 2024) further refines RAG by incorporating reflection tokens labeled by GPT-4 within the instruction-tuning data, enabling the model to better utilize retrieved information. Additionally, we considered CRAG and Self-CRAG(Your et al., 2024), a recent approach that augments standard RAG with corrective strategies to improve retrieval quality by addressing low-quality retrieval results.

## 5 ANALYSIS

In this section, we evaluate the performance of ChunkRAG against existing retrieval-augmented generation (RAG) methods.

### 5.1 COMPARISON

As depicted in Table 1, our method outperformed existing baselines with 64.9% accuracy on PopQA, 77.3% accuracy on PubHealth and 86.4% factscore on Biography when based on *SelfRAG-LLaMA2-7b*.

### 5.2 INSIGHTS

The improvement attained with our technique is mainly due to **chunk-level filtering** and **fine-grained relevance assessment**. We divided the text into semantically meaningful chunks, which reduced the generation of irrelevant or weakly related information. The generation of factually accurate and coherent responses was significantly enhanced due to the filtering mechanism. Notably, chunk-level filtering offers greater benefits in short, fact-intensive tasks like PopQA—where even minor irrelevant

Table 1: Performance Comparison Across Methods: Accuracy on PopQA and PubHealth, and FactScore on Biography. The table summarizes results for LLMs without retrieval, standard RAG approaches, and advanced RAG methods (including ChunkRAG), highlighting improvements in response accuracy and factual consistency.

| Method | PopQA | PubHealth | Biography |
|---|---|---|---|
| **(A) LLMs Without Retrieval** | | | |
| LLaMA2-7B | 14.7 | 34.2 | 44.5 |
| Alpaca-7B | 23.6 | 49.8 | 45.8 |
| LLaMA2-13B | 14.7 | 29.4 | 53.4 |
| Alpaca-13B | 24.4 | 55.5 | 50.2 |
| ChatGPT | 29.3 | 70.1 | 71.8 |
| LLaMA2-chat13B | 20.0 | 49.4 | 55.9 |
| **(B) Standard RAG with LLMs** | | | |
| RAG + LLaMA2-7B | 38.2 | 30.0 | 78.0 |
| RAG + Alpaca-7B | 46.7 | 40.2 | 76.6 |
| RAG + LLaMA2-13B | 45.7 | 30.2 | 77.5 |
| RAG + Alpaca-13B | 46.1 | 51.1 | 77.7 |
| **(C) Advanced RAG** (*SelfRAG-LLaMA2-7b*) | | | |
| RAG | 52.8 | 39.0 | 59.2 |
| Self-RAG | 54.9 | 72.4 | 81.2 |
| CRAG | 59.8 | 75.6 | 74.1 |
| Self-CRAG | 61.8 | 74.8 | 86.2 |
| **ChunkRAG** | **64.9** | **77.3** | **86.4** |

segments can lead to hallucinations—than in open-ended tasks like Biography, which require broader context and thus benefit less from such targeted filtering.

Moreover, the **self-reflective LLM scoring** method, in which the model grades itself and then changes accordingly, led to a significant decrease in retrieval errors. Unlike regular retrieval methods that do not have a filtering mechanism at the document section level, our method can extract more meaningful and relevant information that directly affects the reliability of the generated responses.

# 6 Ablation Studies and Performance Analysis

## 6.1 Redundancy Filtering Effectiveness

To understand the impact of redundancy filtering, we conducted experiments to measure chunk reduction at varying similarity thresholds. Figure 3 demonstrate the percentage reduction in chunks as a function of the similarity threshold, showcasing how filtering removes redundant information. At a threshold of 0.5, the system achieves the highest reduction (20.5%), while more conservative thresholds (e.g., 0.9) reduce the chunks by 8.5%. This analysis provides evidence that redundancy filtering plays a pivotal role in streamlining the retrieval process, significantly reducing irrelevant data.

## 6.2 Performance Without Redundancy Filtering

To gauge the effect of redundancy filtering, we compared the performance of the system with and without filtering. Figure 4 highlights a consistent increase in similarity after filtering, underscoring the improved relevance of retained chunks. Without redundancy filtering, the model frequently integrates irrelevant or loosely related content, leading to degraded relevance scores and higher hallucination rates.

## 6.3 Chunk Merging and Length Analysis

Chunks are dynamically merged based on cosine similarity as part of semantic chunking. Table 2 provides a detailed summary of the number of chunks removed, average chunk length, and the

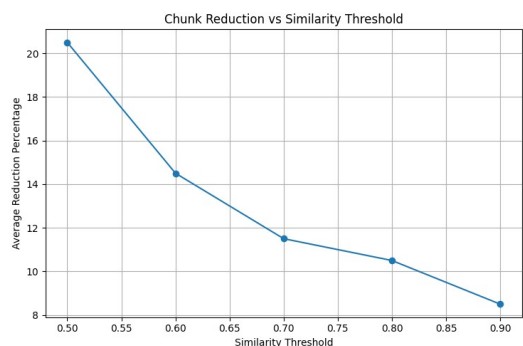

Figure 3: Chunk Reduction vs. Similarity Threshold

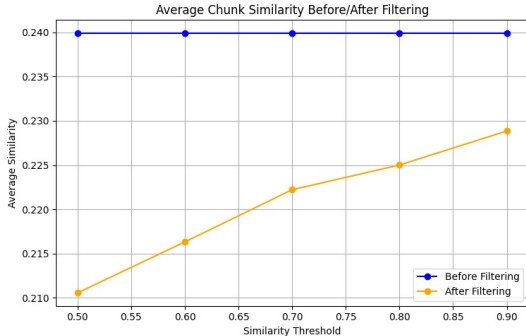

Figure 4: Average Chunk Similarity Before/After Filtering

resultant reduction percentage across different thresholds. For example, at $\theta = 0.6$, 24 chunks are removed, resulting in an average chunk length of approximately 35.66. This adaptive merging mechanism ensures that the retained information remains coherent while minimizing redundancy.

Table 2: Chunk Analysis Across Similarity Thresholds

| Threshold | Chunks Removed | Avg. Length | Similarity Before | Similarity After |
|---|---|---|---|---|
| 0.5 | 36 | 36.04 | 0.2399 | 0.2105 |
| 0.6 | 24 | 35.66 | 0.2399 | 0.2163 |
| 0.7 | 18 | 35.78 | 0.2399 | 0.2222 |
| 0.8 | 16 | 35.49 | 0.2399 | 0.2255 |
| 0.9 | 12 | 35.55 | 0.2399 | 0.2288 |

## 6.4 COMPARATIVE PERFORMANCE ANALYSIS

Table 3 illustrates the performance disparity between the naive retriever and ChunkRAG with $\theta = 0.8$. ChunkRAG consistently outperforms naive retrieval by a significant margin. This highlights the importance of advanced filtering, chunk-level relevance scoring and semantic chunking in improving the retrieval system's effectiveness.

## 7 DISCUSSION

The ablation study highlights redundancy filtering's key role in ChunkRAG, with dynamic chunk merging and optimal similarity thresholds (validated at $\theta = 0.8$) balancing chunk reduction and relevance while preventing over-filtering. Future work could investigate domain-specific thresholds for varying chunk granularity needs and incorporate computational efficiency metrics to assess scalability.

Table 3: Retriever Performance Comparison: Naive Retriever vs. ChunkRAG ($\theta = 0.8$).

| Retriever Type | Average Relevance Score |
|---|---|
| Naive Retriever | 0.180 |
| ChunkRAG ($\theta = 0.8$) | 0.467 |

## 8 CONCLUSION

We introduced ChunkRAG, an LLM-driven chunk filtering method that enhances retrieval-augmented generation precision and factuality through dynamic greedy chunk aggregation. Experiments on PopQA, PubHealth and Biography showed superiority over baselines, with its filtering ensuring relevant, factual chunks were retained during generation, boosting reliability/accuracy and reducing hallucinations in multi-hop tasks. ChunkRAG addresses core LLM retrieval challenges caused by irrelevant or hallucinated content.

## 9 LIMITATIONS

ChunkRAG's effectiveness depends on proper chunk segmentation and embedding quality, as errors can degrade output quality. While successful on PopQA, PubHealth and Biography the system faces challenges including high computational costs from multi-level LLM evaluations and slower processing times due to GPU constraints. Future work could address these limitations through higher-performance GPUs or distributed computing.

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

## A APPENDIX

### A.1 SPECIFIC PROMPTS

#### QUERY REWRITING PROMPT

This prompt refines user queries to better match the underlying documents.

```
You are an AI assistant that improves user queries for better search results.
Rewrite the following query to be more effective for document retrieval without
    changing its meaning.

Original Query: "{query}"

Rewritten Query:
```

#### RELEVANCE SCORING PROMPT

This prompt evaluates the relevance of a text chunk relative to a user query, outputting a single decimal number between 0 and 1.

```
You are an AI assistant tasked with determining the relevance of a text chunk to a
    user query.
Analyze the provided chunk and query, then assign a relevance score between 0 and 1,
     where 1 means highly relevant and 0 means not relevant at all.

Chunk: {chunk}

User Query: {query}

A single decimal number between 0 and 1, representing the final relevance score. No
    other text.

Relevance Score (between 0 and 1):
```

#### SELF-REFLECTION PROMPT

After the initial score is generated, this prompt asks the LLM to reflect on its scoring and adjust if necessary.

```
You have assigned a relevance score to a text chunk based on a user query.
Your initial score was: {score}

Reflect on your scoring and adjust the score if necessary. Provide the final score.

Chunk: {chunk}

User Query: {query}

A single decimal number between 0 and 1, representing the final relevance score. No
    other text.
Final Relevance Score (between 0 and 1):
```

THRESHOLD DETERMINATION PROMPT

This prompt collects the individual relevance scores from various chunks and determines an optimal filtering threshold.

```
Based on the user query and the following set of relevance scores, determine the
    optimal threshold to filter out irrelevant chunks.

Relevance Scores: {scores}

A single decimal number between 0 and 1, representing the final relevance score. No
    other text.
Provide the optimal threshold (between 0 and 1):
```

## A.2 EXAMPLE

**User Query:** "What is Henry Feilden's occupation?"

**Pipeline Steps:**

1. **Query Rewriting:**
   The query is refined from "What is Henry Feilden's occupation?" to "Henry Feilden occupation details biography" to target relevant documents more precisely.

2. **Retrieval:**
   Using the refined query, the system retrieves several text chunks, such as passages containing Henry Feilden's biographical details.

3. **Redundancy Filtering:**
   Overlapping chunks are eliminated to ensure only unique, informative content is retained.

4. **Relevance Scoring:**
   Each chunk is evaluated for its relevance to the query (e.g., a chunk stating "Henry Feilden was a prominent industrialist..." scores high) and its score is fine-tuned if needed.

5. **Thresholding:**
   A dynamic threshold is determined, and only chunks with scores above this value are kept.

6. **Final Output:**
   The remaining chunks are combined to form the final response: *"Henry Feilden is a prominent industrialist, as detailed in his biography."*

