# OpenReview forum: "ChunkRAG: A Novel LLM-Chunk Filtering Method for RAG Systems"
_ICLR.cc/2025/Workshop/BuildingTrust — Submitted to BuildingTrust_

### Official Review · Reviewer_nCtm · 2025-02-23
**Innovative but unclear metrics and model details**

**Rating:** 4
**Confidence:** 3

**Review:**

**Strengths:**
1. The idea of chunk-level filtering addresses a granularity gap in RAG systems.
2. The multi-stage filtering process, integrating redundancy removal, relevance scoring, and hybrid retrieval, is a thorough attempt to refine retrieved content.

**Weaknesses:**
1. Computational efficiency metrics (e.g., runtime) should be included.
2. The critic model refinements are not clearly explained.
3. The dynamic threshold is based on score distribution but lacks a clear rationale or comparison to static thresholds.
4. The paper claims to address the "Lost in the Middle" problem via Cohere’s reranking model but provides no ablation.

**Questions:**
1. Table 2, "Chunk Analysis Across Similarity Thresholds". What do these similarities measure? (inter-chunk similarity or query similarity)

---

### Official Review · Reviewer_TB6D · 2025-03-02

**Rating:** 5
**Confidence:** 3

**Review:**

### Summary
This paper introduces ChunkRAG, an approach to improving RAG systems by implementing chunk-level filtering. The authors propose a dynamic greedy chunk aggregation strategy that segments documents into semantically coherent, variable-length sections based on cosine similarity. The approach is evaluated on several datasets (PopQA, PubHealth, and Biography).

### Pros

The paper addresses an important issue in RAG systems, on chunking long files. The authors conduct experiment on multiple datasets (PopQA, PubHealth, and Biography) and conduct analysis on the chunk reduction.

### Cons
1. One important baseline is other ways to chunk the documents. In many works, they chunk documents by sentences/paragraphs, or simply 256 words per chunk, and it works very well. Such simple chunking methods have not been compared in this paper, and I doubt what is the improvement of “semantic chunking” gives compared with these naive methods.
2. The paper would be benefited from evaluation on more rag tasks, with more diversity on topic and document lengths. For example, the widely used mteb tasks.

---

### Official Review · Reviewer_knTS · 2025-03-02
**Paper with traditional methods**

**Rating:** 5
**Confidence:** 4

**Review:**

**Summary**
This paper is trying to reduce the mistakes by document-level RAG for LLMs. They proposed a chunk-size RAG based method with initial filtering and multi-stage scoring. Their evaluation shows that the proposed method can outperform other advanced RAG systems.

**Strengths**
  1. The paper is clear and easy to understand and follow.
  2. The multi-stage scoring idea seems interesting.

**Weakness**
  1. Chunked RAG is not a new idea. The paper did not talk about the difference between their method with other chunked RAG based methods (e.g., [1]).
  2. All the figures used in the paper should be updated.
  3. Figure 1 cannot clearly show the advantage of chunked RAG, since the left output is also acceptable for human beings, as human beings prefer longer answers with more details.
  4. The experiments did not show why the threshold for similarity $\theta=0.8$ is the optimal choice.
  5. I am curious if the initial filtering of redundancy (similarity = 0.9) can do harm or do good to the overall retrieval and the final answer correctness, since although 0.9 seems very high, if remove either chunk, then there is still a great amount of information loss, which might contain the correct answer or critical hints to find the final answer.

[1] Zhong, Zijie, et al. "Mix-of-granularity: Optimize the chunking granularity for retrieval-augmented generation." arXiv preprint arXiv:2406.00456 (2024).

---

### Decision · Program_Chairs · 2025-03-02

Reject